# Regulation of Cellular Senescence Is Independent from Profibrotic Fibroblast-Deposited ECM

**DOI:** 10.3390/cells10071628

**Published:** 2021-06-29

**Authors:** Kaj E. C. Blokland, Habibie Habibie, Theo Borghuis, Greta J. Teitsma, Michael Schuliga, Barbro N. Melgert, Darryl A. Knight, Corry-Anke Brandsma, Simon D. Pouwels, Janette K. Burgess

**Affiliations:** 1University of Groningen, University Medical Center Groningen, Department of Pathology and Medical Biology, 9713 GZ Groningen, The Netherlands; k.e.c.blokland@umcg.nl (K.E.C.B.); t.borghuis@umcg.nl (T.B.); g.j.teitsma@umcg.nl (G.J.T.); b.n.melgert@rug.nl (B.N.M.); c.a.brandsma@umcg.nl (C.-A.B.); s.d.pouwels@umcg.nl (S.D.P.); 2University of Groningen, University Medical Center Groningen, Groningen Research Institute for Asthma and COPD, 9713 GZ Groningen, The Netherlands; h.habibie@rug.nl; 3School of Biomedical Sciences and Pharmacy, University of Newcastle, Callaghan, NSW 2308, Australia; michael.schuliga@newcastle.edu.au (M.S.); dknight2@providencehealth.bc.ca (D.A.K.); 4National Health and Medical Research Council Centre of Research Excellence in Pulmonary Fibrosis, Sydney, NSW 2050, Australia; 5University of Groningen, Department of Molecular Pharmacology, Groningen Research Institute of Pharmacy, 9713 GZ Groningen, The Netherlands; 6Faculty of Pharmacy, Hasanuddin University, Makassar 90245, Indonesia; 7Department of Anesthesiology, Pharmacology and Therapeutics, University of British Columbia, Providence Health Care Research Institute, Vancouver, BC V6Z 1Y6, Canada; 8Department of Pulmonology, University Medical Center Groningen, University of Groningen, 9713 GZ Groningen, The Netherlands

**Keywords:** extracellular matrix, senescence, idiopathic pulmonary fibrosis, proinflammatory, profibrotic

## Abstract

Idiopathic pulmonary fibrosis (IPF) is a devastating lung disease with poor survival. Age is a major risk factor, and both alveolar epithelial cells and lung fibroblasts in this disease exhibit features of cellular senescence, a hallmark of ageing. Accumulation of fibrotic extracellular matrix (ECM) is a core feature of IPF and is likely to affect cell function. We hypothesize that aberrant ECM deposition augments fibroblast senescence, creating a perpetuating cycle favouring disease progression. In this study, primary lung fibroblasts were cultured on control and IPF-derived ECM from fibroblasts pretreated with or without profibrotic and prosenescent stimuli, and markers of senescence, fibrosis-associated gene expression and secretion of cytokines were measured. Untreated ECM derived from control or IPF fibroblasts had no effect on the main marker of senescence p16^Ink4a^ and p21^Waf1/Cip1^. However, the expression of alpha smooth muscle actin (ACTA2) and proteoglycan decorin (DCN) increased in response to IPF-derived ECM. Production of the proinflammatory cytokines C-X-C Motif Chemokine Ligand 8 (CXCL8) by lung fibroblasts was upregulated in response to senescent and profibrotic-derived ECM. Finally, the profibrotic cytokines transforming growth factor β1 (TGF-β1) and connective tissue growth factor (CTGF) were upregulated in response to both senescent- and profibrotic-derived ECM. In summary, ECM deposited by IPF fibroblasts does not induce cellular senescence, while there is upregulation of proinflammatory and profibrotic cytokines and differentiation into a myofibroblast phenotype in response to senescent- and profibrotic-derived ECM, which may contribute to progression of fibrosis in IPF.

## 1. Introduction

Idiopathic pulmonary fibrosis (IPF) is a devastating interstitial lung disease of unknown aetiology that is characterised by exaggerated deposition of extracellular matrix (ECM) leading to an irreversible decline in lung function and ultimately death [1]. Genetics, gender and ageing, as well as external factors such as exposure to cigarette smoke or air pollution, have been identified as risk factors for IPF. The disease presents mainly in the elderly, with a mean survival rate of 2–5 years after diagnosis. Ageing is characterised as functional decline of tissue over time leading to dysfunction. Changes in the ECM of the lung during ageing, such as loss of elastin and increased collagen contributes to aberrant composition that negatively influences proliferation, migration and differentiation of cells [2]. Mesenchymal cells, such as residing fibroblasts, are responsible for maintaining and remodelling the ECM during homeostasis. In IPF, both alveolar epithelial cells and fibroblasts show signs of growth arrest and resistance to apoptosis, which are indicative of cellular senescence, a hallmark of ageing [3,4,5]. Cellular senescence is also characterised by the induction of a hypersecretory phenotype known as the senescence-associated secretory phenotype (SASP) [6,7,8]. It is now well recognised that cellular senescence contributes to the pathogenesis of lung fibrosis [5,9]. While senescence is an important regulatory element during wound repair, dysregulation and accumulation of senescent cells during ageing contributes to an aberrant repair response that leads to fibrosis [6,10]. The negative impact of senescence is mainly attributed to SASP and the inability to clear senescent cells favouring disease progression.

In IPF, the accumulation of fibroblasts and ECM leads to formation of fibroblastic foci, the active regions of fibrosis in which myofibroblasts are responsible for excessive matrix deposition and that invade morphologically healthy lung [11]. The changes that occur to the ECM drastically influence the composition, topography, stiffness and biomechanics of lung tissue [12]. These changes are recognised as a major driver of aberrant cellular responses in IPF [13,14]. The pathological increase in ECM deposition and stiffness causes a mechanical gradient between normal and fibrotic tissue, impacting proliferation, migration and differentiation [15]. In recent years, more attention has been directed towards the exact role of ECM in disease progression and how individual ECM proteins may directly impact cellular function. ECM proteins, such as laminins and collagens, are differentially expressed in fibrotic and senescent cells [16,17]. However, the contribution of aberrant ECM in the induction of cellular senescence in fibrotic diseases is underexplored. Evidence is emerging that several core proteins such as collagen, elastin and fibronectin may contribute to pathological senescence [18,19,20]. Increased collagen and decreased elastin deposition contribute to a stiffer ECM and subsequent TGF-β activation, the main profibrotic cytokine. Fibronectin is important for collagen fibril formation and is known to be dysregulated in IPF. Although the exact role of ECM in relation to cellular senescence is unknown, the ability to bind several growth factors such as platelet-derived growth factor (PDGF) and fibroblast growth factor (FGF) may contribute to senescence induction [21,22]. Another group of ECM proteins known as matricellular proteins do not directly contribute as major structural proteins but are secreted in the extracellular environment. Decorin (DCN), fibulin1 (FBLN1) and connective tissue growth factor (CTGF) have been implicated in fibrosis, and there is evidence suggesting a role in cellular senescence as well [23,24,25].

We hypothesised that the altered composition of fibrotic ECM negatively impacts fibroblast function and leads to cellular senescence in IPF. Here, we aimed to investigate whether IPF-derived ECM was able to induce cellular senescence by culturing primary control fibroblasts on Ctrl– and IPF–fibroblast or senescent and profibrotic-derived decellularized ECM and, subsequently, analyse gene expression of markers of senescence and fibrosis and secreted factors that are both part of the SASP and the fibrotic response.

## 2. Materials and Methods

### 2.1. Human Lung Tissue and Culture

Primary human lung fibroblasts were obtained from lung tissue of patients undergoing lung transplantation or tumour resection surgery at the University Medical Center Groningen (UMCG). The protocol was consistent with the Research Code (https://www.umcg.nl/EN/Research/Researchers/General/ResearchCode/Paginas/default.aspx; last accessed 28 May 2021) and national ethical and professional guidelines (“Code of conduct; Dutch federation of biomedical scientific societies,” https://www.federa.org; last accessed 28 May 2021). Additional lung fibroblasts were obtained from the Woolcock Institute of Medical Research, Australia. Approval was provided by the Human Ethics Committee of the University of Sydney and the Sydney South West Area Health Service. Fibroblasts were isolated from macroscopically and histologically normal tissue as described before [26,27].

Primary lung fibroblasts were cultured in low glucose Dulbecco’s Modified Eagle Medium (DMEM) (Lonza, Geleen, The Netherlands) supplemented with 10% foetal bovine serum (FBS; Sigma–Aldrich, Zwijndrecht, The Netherlands), 100 U/mL penicillin, 100 µg/mL streptomycin (Gibco, Breda, The Netherlands) and 1% GlutaMAX (Gibco), referred to as complete growth medium, and used between passage 4 and 6. Fibroblasts were routinely checked for mycoplasma infection using a PCR assay and used only when certified negative [28]. To quiesce, fibroblasts were cultured for 24 h in low glucose DMEM containing 0.1% bovine serum albumin (BSA; Sigma–Aldrich), referred to as incomplete growth media.

Table 1 shows the characteristics of the ECM fibroblast donors used in this study. All donors were age and sex matched where possible.

### 2.2. ECM Generation

Fibroblasts (14,000 cells/cm^2^) were seeded per well in complete growth media and allowed to attach for 24 h. Cells were then maintained in incomplete growth media for another 24 h to quiesce the fibroblasts before treatment. For the generation of profibrotic ECM, fibroblasts were stimulated with 1 ng/mL transforming growth factor β1 (TGF-β1; Gibco #PHG9214) for 72 h. To induce cellular senescence and senescent ECM, fibroblasts were exposed to 150 µM hydrogen peroxide (H_2_O_2_; Sigma–Aldrich) for 2 hrs, washed twice with an equal amount of phosphate buffered saline (PBS; Gibco) and incubated in complete growth media containing 5% FBS for 72 h [29]. An untreated control group was generated using Ctrl–fibroblasts and IPF–fibroblasts grown in complete media for 72 h and a negative no-ECM control was generated by “coating” the culture plate with complete growth media.

Decellularization of fibroblast cultures were performed using 20 mM ammonium hydroxide in ddH_2_O [30]. Supernatant was aspirated and stored at −80 °C until further analysis. First, cell cultures were washed with PBS followed by 5 min incubation at room temperature in 20 mM ammonia water until cells were no longer visible on an Olympus CK2 microscope. Then, ammonia water was aspirated, and cell-free cultures were washed three times in equal amounts of PBS before being stored at −20 °C in PBS until further usage. This resulted in cell-free and intact ECM, as confirmed by scanning electron microscopy (SEM; Appendix A).

### 2.3. Scanning Electron Microscopy (SEM)

Fibroblasts (14,000 cells/cm^2^) were seeded on cover slips in complete growth medium and underwent the same treatment as described above. Coverslips were decellularized and fixed in 2% PFA and 0.2% glutaraldehyde and prepared for SEM. Cover slips were washed with 0.1 M cacodylate buffer and incubated with 1% osmium tetroxide in 0.1 M cacodylate buffer for 1 h at room temperature. Then, cover slipes were washed with water and dehydrated with 30, 50 and 75% ethanol (EtOH), 15 min per each step, respectively, followed by dehydration in 100% EtOH for 30 min. To dry the samples, cover slips were first incubated in a 1:1 mixture of 100% EtOH and tetramethyl silane (TMS) for 10 min followed by 15 min incubation with 100% TMS after which the cover slips were air dried. Afterwards cover slips were gold coated using a sputter coater and cover slips were imaged using a Zeiss Supra55 at 3 KV with SE2 detector (Carl Zeiss, Breda, The Netherlands).

### 2.4. Culture of Fibroblasts on ECM Substrate

Culture plates with ECM substrate were thawed at 37 °C and rinsed once with sterile PBS. PBS was aspirated, and two different fibroblast donors (Table 2) were seeded (14,000 cells/cm^2^) back on the ECM substrate in complete growth media and cultured for 72 h. After 72 h, culture supernatants were collected, pooled and centrifugated for 5 min at 300× *g* to remove debris and dead cells, after which cell-free supernatant was stored at −80 °C for further analysis. Fibroblasts were incubated using 0.5% trypsin/EDTA (Sigma–Aldrich) until all cells were detached. The trypsin/EDTA mixture was inactivated using Hank’s Balanced Salt Solution (HBSS; Sigma–Aldrich) containing 5mM ethylenediaminetetraacetic acid (EDTA) and 2.5% *v*/*v* FBS. Cells were stained with trypan blue (Sigma–Aldrich) and manually counted using a haemocytometer. In addition, cells were lysed in RNA lysis buffer (Machery-Nagel, Düren, Germany) and stored at −80 °C until RNA isolation. Cells to be used for SA-β-Gal staining were fixed with 2% paraformaldehyde (PFA; Sigma–Aldrich) and 0.2% glutaraldehyde in PBS for 5 min and directly stained using the protocol described below.

### 2.5. Gene Expression Analysis

Total RNA was isolated using a Nucleospin RNA isolation kit (Machery-Nagel) according to manufacturer’s instructions, after which the samples were stored at −80 °C. Total RNA was quantified using the Nanodrop ND−2000 (Thermo Scientific, Breda, The Netherlands). RNA was reversed transcribed into cDNA using the ReverseAid First Strand cDNA Synthesis Kit (Thermo Scientific). DNA was amplified using the GoTaq Probe qPCR Master Mix (Promega, Leiden, The Netherlands) and TaqMan Gene Expression Assay (Thermo Scientific) on a ViiA7 Real-Time PCR System (Applied Biosystems, Breda, The Netherlands), with the relevant predeveloped primers listed in Table 3 (Thermo Scientific) in a duplex reaction for both gene of interest and reference gene in the same well. Data were extracted using QuantStudioTM 6 software (Applied Biosystems) and processed using the following approach: Delta (d)Ct was calculated by subtracting the Ct values of the 18S reference gene from those of the gene of interest (GOI) and then transformed to 2-dCt for analysis. Values were excluded when the difference between the triplicate values for any gene were >0.2 Ct for 18S and >0.5 Ct for GOI. The limit of quantification for all primers was set at a Ct threshold of 35.

### 2.6. Senescence-Associated β-Galactosidase Staining

Cellular senescence was assessed using a previously described protocol for senescence-associated β-galactosidase (SA-β-Gal) staining [31]. Fixed fibroblasts were washed with PBS, and SA-β-Gal staining solution was added. Plates were incubated in a dry incubator at 37 °C for 16 h. Staining solution was aspirated before being washed with PBS. PBS containing 1 µg/mL 4′,6-diamidino-2-phenylindole (DAPI; Sigma–Aldrich) was added, and plates were incubated in the dark for 10 min, washed and stored in 70% glycerol in PBS at 4 °C. Plates were imaged using a TissueFAXS automated analysis system (TissueGnostics, Vienna, Austria). Brightfield (SA-β-Gal) and DAPI images were exported using TissueFAXS Viewer 7.0 (TissueGnostics) and analysed in FIJI [32,33]. Total cell numbers and SA-β-Gal positive cells were used to calculate the percentage of SA-β-Gal positive cells.

### 2.7. Enzyme Linked Immunosorbent Assay (ELISA)

Secreted IL-6, CXCL8, DCN, TGF-β1, Osteoprotegerin (OPG), receptor activator of nuclear factor kappa-B ligand (RANKL) and CTGF were measured using Human DuoSet ELISA kits (respectively cat#DY206, cat#DY208, cat#DY143, cat#DY240, cat#DY805, cat#DY626, cat#DY9190-05; R&D Systems, Abingdon, UK) according to the manufacturer’s instructions. Colour development was obtained through the reaction between 3,3′, 5,5″-tetramethylbenzidine (TMB) and horseradish peroxidase (HRP). As all extracellular RANKL, excreted by cells, was complexed to OPG and not detectable with a direct RANKL ELISA, a RANKL/OPG complex sandwich ELISA was developed by using a human capture RANKL antibody (part of cat#DY626, R&D Systems) and a human detection OPG antibody (part of cat#DY805, R&D Systems) [26]. The total concentrations of RANKL and OPG were determined by adding the free and complex-bound protein concentration. To normalise ELISA data, cell numbers were used to express protein concentration per 10^5^ cells counted.

### 2.8. Data and Statistical Analyses

RT-qPCR data were analysed for outliers using the following approach. First, the coefficient of variability (CV) was calculated for 18S and GOI. Then, technical triplicates with a %CV above 1 were inspected and excluded if 18S was >0.2 Ct and >0.5 Ct for GOI as technical errors. SA-β-Gal outliers were detected using the following approach. The variation between duplicate samples and the number of positive cells in ImageJ did not match with the TissueFAXS images. Statistical analyses were performed using GraphPad Prism 8 (GraphPad software, San Diego, CA, USA). Data were tested for normal distribution using the D’Agostino and Pearson test. Data were log-transformed if distribution was not normally distributed, after which all data was normally distributed; then, we used a repeated measures one-way ANOVA, or, if data points were missing, a mixed-effects analysis (REML) was used. ANOVA tables can be found in the Appendix A. Wilcoxon matched-pairs signed rank test was used for Ctrl versus IPF–ECM. A post hoc Sidak multiple comparisons test was used for differences within ECM treatment groups. All statistical analyses were performed using the raw data. Data were considered significantly different at *p* < 0.05. Significance was either expressed as *p*-value or * *p* < 0.05, ** *p* < 0.01 and *** *p* < 0.001 unless stated otherwise.

## 3. Results

### 3.1. ECM Deposition by Control and IPF Lung Fibroblasts

To investigate the effect of deposited ECM by control or IPF lung fibroblasts on the senescence induction, we first wanted to confirm that our decellularization protocol resulted in cell-free matrix deposition. Figure 1 shows the deposited ECM on glass cover slips from one control and one IPF donor. Under all conditions, there is ECM deposition. Fibroblast treated with TGF-β1 show increased total ECM deposition and structure, of which IPF-derived demonstrates the most change. Interestingly, ECM deposition from senescent fibroblasts is much less, with smaller differences compared to untreated.

### 3.2. IPF–ECM Does Not Modulate Markers of Senescence in Primary Human Lung Fibroblasts

To determine if IPF-derived ECM predisposes control fibroblasts to a senescent phenotype, we examined several markers of senescence in two fibroblast donors cultured on Ctrl- and IPF-derived ECM from six donors per group. Data generated using donor one is shown in the main manuscript, while data from donor two is presented in the Appendix A. IPF-derived ECM did not alter proliferation or the number of SA-β-Gal-positive cells in fibroblasts compared to Ctrl-derived ECM after 72 h of culture in (Figure 2A,B). No morphological differences were observed between the fibroblasts cultured on Ctrl- or IPF-derived ECMs. SA-β-Gal images that are representative of six unique ECM donors are shown in Appendix A. Analysis of gene expression of cell-cycle inhibitors p21^Waf1/Cip1^ and p16^Ink4a^ confirmed this finding, as there was no change in expression in response between Ctrl- and IPF-derived ECM in both fibroblast donors (Figure 2C,D). Next, gene expression and protein secretion of the SASP factors IL-6 and CXCL8 was measured but did not change in fibroblasts cultured on IPF-derived ECM compared to Ctrl-derived ECM (Figure 2E–H). Donor two shown in Appendix A demonstrates a similar response to IPF-derived ECM.

### 3.3. Upregulation of α-SMA and DCN Gene Expression in Fibroblasts

IPF-derived ECM did not alter markers of cellular senescence. We therefore sought to characterise if IPF-derived ECM would activate and/or modulate a profibrotic response. We measured gene expression and secreted proteins that are identified to be part of the fibrotic response in IPF. Gene expression analysis demonstrated that ACTA2 gene expression was higher in fibroblasts when cultured on IPF-derived ECM, when compared to Ctrl-derived ECM (Figure 3A). However, there was no change in expression of COL1A1, FN1 and DCN on IPF-derived ECM (Figure 3B–D). Analyses of secreted DCN demonstrated an increase in fibroblasts on IPF-derived ECM. Secretion senescence (RANKL) and fibrosis-associated factors (OPG, TGF-β1 and CTGF) in fibroblasts did not demonstrate a change in response to IPF-derived ECM (Figure 3E–I). FBLN1c gene expression did not change in response to IPF-derived ECM (Appendix A). Furthermore, while TGF-β1 secretion was not altered, there was an increase in gene expression (*p* = 0.04) on IPF-derived ECM (Appendix A). To further test if IPF-derived ECM would promote its own cross-linking, we analysed TG2, LOX and LOXL 1–4. Analyses demonstrated that fibroblasts cultured for three days on IPF-derived ECM did not change expression of TG2, LOX and LOXL1–4 when compared to Ctrl-derived ECM (Appendix A). Donor two is shown in Appendix A, which demonstrate a similar pattern of expression for all targets except TGF-β1. Secretion of TGF-β1 is upregulated (*p* = 0.046) in fibroblasts cultured on IPF-derived ECM when compared to Ctrl-derived ECM (Appendix A). Gene expression of TGF-β1 is not altered in response to IPF-derived ECM (Appendix A).

### 3.4. Fibrotic ECM Activates a Secretory Phenotype in Primary Lung Fibroblasts

As fibroblasts cultured on IPF-derived ECM do not become senescent, we investigated whether ECM derived from senescence-induced fibroblasts would have a modulatory effect. Fibroblasts were cultured for three days, and markers of senescence were assessed as described before. ECM derived from fibroblasts treated with either H_2_O_2_ or TGF-β1 did not change fibroblasts cell number or SA-β-Gal detection (Appendix A) after three days of culture when compared to untreated Ctrl fibroblasts-derived ECM (Figure 4A,B). No morphological changes were observed in fibroblasts cultured on treatment-derived ECM. Furthermore, no changes in cell-cycle inhibitors p21^Waf1/Cip1^ and p16^Ink4a^ were observed after three days of culture on treated ECM (Figure 4C,D). Analysis of both gene expression and secretion of IL-6 and CXCL8 demonstrated no difference in gene expression in response to ECM derived from Ctrl or IPF fibroblasts (Figure 4E–H). CXCL8 was higher on ECM derived from TGF-β1-treated IPF fibroblasts when compared to ECM derived from nontreated IPF fibroblasts (Figure 4H). Donor two is shown in Appendix A, which demonstrates a similar pattern of expression in proliferation, SA-β-Gal cytochemical staining and gene expression. However, there is no difference in secretion of IL-6 in response to treatment-derived ECM, while CXCL8 only shows an increase on TGF-β1 treatment of IPF-derived ECM after three days in culture (Appendix A).

### 3.5. Higher Secretion of TGF-β and CTGF in Response to Fibrotic and Senescent Fibroblast Derived ECM

Fibrotic- and senescence-derived ECM modulated the SASP cytokines IL-6 and CXCL8 in fibroblasts. To further investigate the response to stimulated ECM, we characterised fibrosis-associated gene expression and cytokine production in fibroblasts. The expression of ACTA2, COL1A1, FN1 and DCN did not demonstrate a difference in response to fibrotic- or senescence-derived ECM (Figure 5A–D). Analyses of FBLN1C and TGF-β1 expression did not change either in response to treatment-derived ECM (Appendix A). Analysis of cross-linking enzymes TG2, LOX and LOXL1–4 also did not show any response towards treatment-derived ECM (Appendix A). Finally, we measured secretion of several cytokines that are part of the fibrotic response. Secretion of DCN, RANKL and OPG did not change in response to treatment-derived ECM in fibroblasts (Figure 5E–H). However, there was increased secretion of TGF-β1 and CTGF. TGF-β1 demonstrated an increase (*p* = 0.003, *n* = 6) on senescent-derived Ctrl-ECM compared to control (Figure 5H). CTGF showed an increase (*p* = 0.002, *n* = 6) on senescent derived Ctrl-ECM, while fibrotic-derived IPF–ECM demonstrated a decrease (*p* = 0.02, *n* = 6) in fibroblasts after three days of culture (Figure 5I). Donor two is shown in Appendix A, which illustrate a similar expression profile for all genes measured. The secretion of TGF-β1 and CTGF were increased for both fibrotic and senescent-derived Ctrl–ECM (Appendix A).

## 4. Discussion

In IPF, both cellular senescence and excessive/altered ECM deposition are identified as major drivers of fibrosis. These changes impact cell function and contribute to disease progression by creating a positive feedback loop favouring fibrosis. In the present study, we cultured fibroblasts on Ctrl- or IPF-fibroblast-derived ECM for three days and examined the cellular response, focusing on cellular senescence and expression of fibrotic markers. We report that unstimulated Ctrl- or IPF-derived ECM did not affect markers of senescence, such as p16^Ink4a^, p21^Waf1/Cip1^, or a series of secreted factors, associated both senescence and fibrotic responses. However, when fibroblasts were cultured on ECM derived from TGF-β1- or H_2_O_2_-stimulated Ctrl- or IPF-fibroblasts, we detected upregulation in CXCL8, CTGF and TGF-β1. Collectively, the data presented in this study suggest that exposure of fibroblasts to Ctrl- or IPF-derived ECM under basal conditions does not induce cellular senescence nor induce cytokine production. Importantly, fibroblasts exposed to stimulated active environment differentially secrete factors that are likely to impact further fibrotic activities.

Ctrl- and IPF-derived ECM generated from fibroblasts under basal and stimulated conditions which were decellularized and ECM deposition was confirmed with SEM. These ECMs displayed differences in structure between the basal and treatment groups. However, owing to technical limitations with the SEM, the images were captured of ECMs deposited on glass cover slips, which may reflect an underestimation of the amount of ECM deposited upon tissue culture plastic, as used for our cell seeding experiments. Furthermore, the SEM images in this study of cell-free ECM are comparable to those of Philp and colleagues [14]. However, there may be some differences in the quantity of deposited ECM, as the time allowed to deposit ECM varied between the two studies. Once we confirmed that our protocol removed the cells, while leaving ECM proteins intact, we then investigated what was the impact of Ctrl- or IPF-derived ECM on fibroblast senescence. In both donors tested, no induction of senescence was observed when fibroblasts were cultured on Ctrl or IPF-derived ECM under basal conditions. This is the first study to show that ECM, by itself, does not induce senescence in fibroblasts. This finding is supported by a recent study in which Faiz and colleagues found that ECM from nonstimulated asthmatic airway smooth muscle (ASM) cells functions as healthy ASM cells [34]. Our findings might suggest that the ECM deposited by fibroblasts under basal conditions does not invoke a microenvironment that influences the senescence state of cells in its immediate vicinity.

Decorin secretion was increased from fibroblasts exposed to IPF-derived ECM, compared to Ctrl-derived ECM. DCN has been implicated in IPF and plays a role in wound repair by limiting the function of both TGF-β1 and CTGF [35]. The fibroblasts were exposed to ECM for three days, which would be expected to be more than sufficient for cells to respond to their environment. Parker and colleagues previously reported that, albeit on decellularized lung slices, fibroblasts cultured on both control and IPF–ECM responded to their environment after 18 h of culture [13]. In addition, in a recent study, the authors demonstrated that reseeded fibroblasts on decellularized lung scaffolds had a temporal response favouring proliferation and adhesion before increasing productions of core matrisome proteins in long term cultures [36]. In concert, these data illustrate the ability of fibroblasts to quickly respond to their environment; however, it is unclear if this response is due to increased stiffness, altered ECM composition or a combination of both in IPF.

The current study was designed to examine the effect of ECM on the induction of a senescent phenotype in fibroblasts. When the ECM was generated under basal conditions, both the control- and IPF-derived ECMs failed to induce changes in the senescence status of the fibroblasts, clearly demonstrating the assuaging effects of the ECM deposited when the fibroblasts are in a nonstimulatory environment. To further investigate the response of fibroblasts, we generated ECM under senescent or fibrotic conditions to mimic active fibrosis in control– and IPF–fibroblasts. Next, we characterised cellular senescence and the fibrotic response. Fibroblasts cultured on senescent- or fibrotic-derived ECM did not show an induction of p16^Ink4a^ and p21^Waf1/Cip1^, the main markers of senescence; instead, it resulted in upregulation of proinflammatory cytokine CXCL8 compared to control ECM. These data indicate that ECM generated under stimulated conditions mimicking senescence or profibrotic environment leads to a cellular response, which is in line with what has been reported previously [14]. Increased secretion of IL-6 and CXCL8, in response to treated ECM, is expected, as both cytokines are important early in wound repair [3]. Together with upregulation of DCN under basal conditions in IPF, these data suggest that there is a wound repair response initiated by the ECM environment.

To further expand on the finding that senescent- or profibrotic-derived ECM leads to activation of proinflammatory cytokines, we characterised fibrosis target genes and secretion profile. Fibroblasts cultured on IPF-derived ECM under stimulated conditions led to increased TGF-β1 and CTGF secretion. Surprisingly, we did not measure higher secretion levels of OPG in response to IPF–ECM. OPG is essential for ECM homeostasis by osteoclasts in the bone. Moreover, OPG is induced by TGF-β1, and it is suggested to bind CTGF, potentially contributing to fibrosis [37]. On the other hand, RANKL has been identified as a part of the SASP in COPD fibroblasts [38]. Given the fact that we see no modulation of senescence, it was not unexpected to see a lack or increase in RANKL secretion. Furthermore, gene expression of cross-linking enzymes was not altered compared to on unstimulated ECM. This is in contrast to a previous study, in which it was reported in IPF that the fibrotic ECM leads to the creation of a feedback loop in fibroblasts, with increased cross-linking of the ECM facilitated by increased production of enzymes responsible for this process [14]. It might be possible that the different exposure time to the ECM between these two studies was enough to generate these alternate responses. Development of IPF usually takes years, and during this time, the cells in the lungs are exposed to a continuous stream of stimuli (signals) that leads to the pathological changes seen in end-stage fibrosis [39]. These changes may go stepwise, for example, pathological stiffness is a product of increased ECM deposition, fibril organisation and, ultimately, enzymatic cross-linking. However, in order to achieve increased stiffness, all three must be present, which, in fact, might take years to accumulate. 

The experimental setup for this study was designed to measure the impact of ECM on the senescence phenotype after three days of culture. We did not observe any modulation in senescence markers after three days. This could implicate, in the context of senescence, that fibroblasts need prolonged exposure to an altered ECM before they make this adaptation. There is evidence suggesting that the upregulation of proinflammatory cytokines may precede the induction of cell-cycle inhibitors p21^Waf1/Cip1^ and p16^Ink4a^ in senescence, as reflected in this study [40]. Moreover, as IPF is a multifactor disease, a secondary stimulant such as reactive oxygen species (ROS) leading to DNA damage, pathological stiffness might be needed to create an environment that favours cellular senescence. Lastly, the origin of fibroblast donors used to culture on top of the ECM in this study may also be a factor in the outcome of the results. IPF is predominantly diagnosed in the elderly, and up to 80% of patients have a history of smoking [41]. It is not unimaginable that the biological response differs between young and older donors or those with smoking history [42].

## 5. Conclusions

In conclusion, we have shown that fibroblast exposed to Ctrl- or IPF-derived ECM under nonstimulated conditions does not induce cellular senescence. Instead, we observed upregulation of proinflammatory cytokines and profibrotic cytokines in response to senescent or profibrotic ECM. These data suggest that, to induce a strong response to their environment, not only is composition important in IPF but a secondary stimulant that induces DNA damage such as ROS, or pathological stiffness might be required to create a positive feedback look that perpetuates fibrosis in IPF.

## Figures and Tables

**Figure 1 cells-10-01628-f001:**
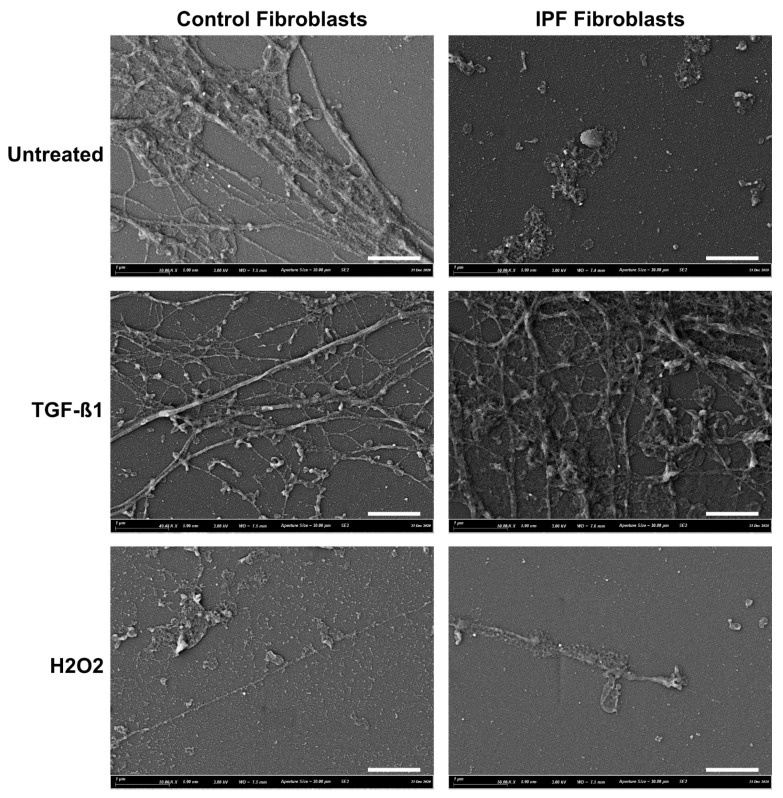
SEM images of deposited ECM from control and IPF fibroblasts. ECM preparation of Ctrl- and IPF-derived ECM under basal (untreated) or stimulated conditions (TGF-β1 and hydrogen peroxide). Images show different cell-free deposited ECM proteins. Images are representative of two donors. Scale bar = 1 µM.

**Figure 2 cells-10-01628-f002:**
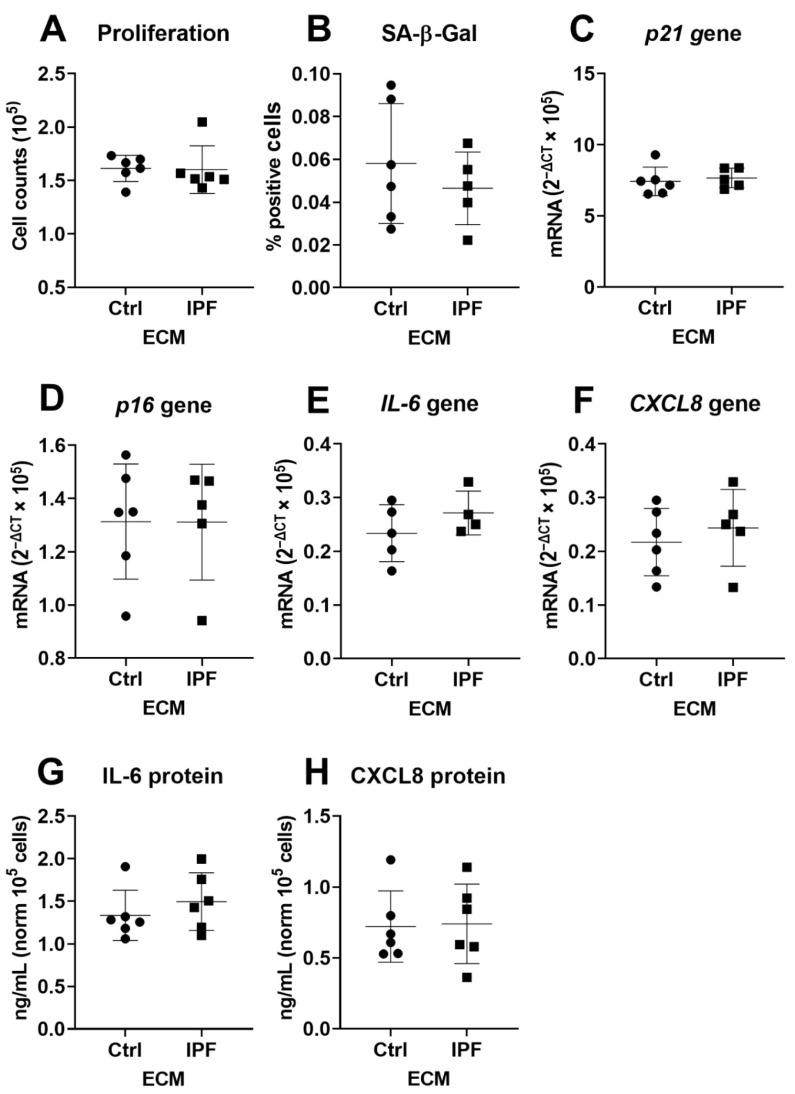
Markers of senescence in fibroblasts cultured on Ctrl- or IPF-derived ECM. Fibroblasts were cultured for up to three days on Ctrl- or IPF-derived ECM, and proliferation was assessed by cell enumeration (**A**), and SA-β-Gal-positive cells were counted (**B**). Panels (**C**,**D**) demonstrate cell-cycle inhibitors p21^Waf1/Cip1^ and p16^Ink4a^ after three days of culture. Panel (**E**–**H**) depicts the gene expression and protein secretion of known SASP factors IL-6 and CXCL8 (*n* = 5–6). Wilcoxon matched-pairs signed rank test was used to measure difference between Ctrl and IPF and was considered significant at *p* < 0.05.

**Figure 3 cells-10-01628-f003:**
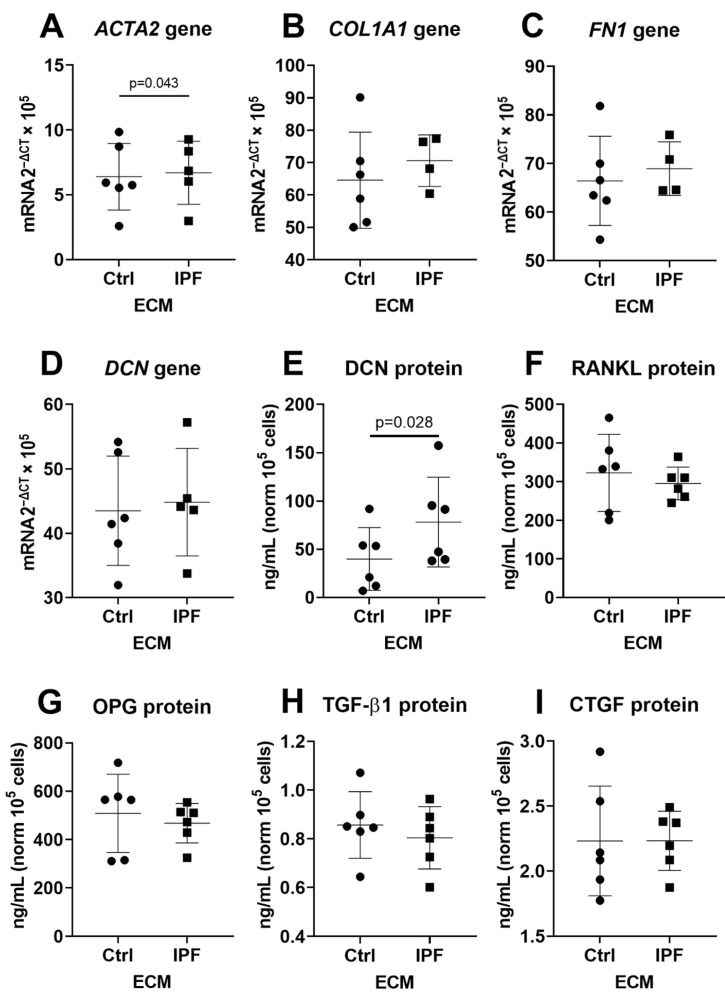
Effect of IPF-derived ECM on the expression of fibrosis-associated genes and secretion of profibrotic cytokines. Fibroblasts were cultured for three days on Ctrl- or IPF-derived ECM before expression levels of ACTA2, COL1A1, FN1 and DCN were assessed (**A**–**D**). The secretion of known fibrotic factors DCN, RANKL, OPG, TGF-β1 and CTGF was also evaluated (**E**–**I**) (*n* = 4–6). Wilcoxon matched pairs signed rank test was used to measure difference between Ctrl and IPF and was considered significant at *p* < 0.05.

**Figure 4 cells-10-01628-f004:**
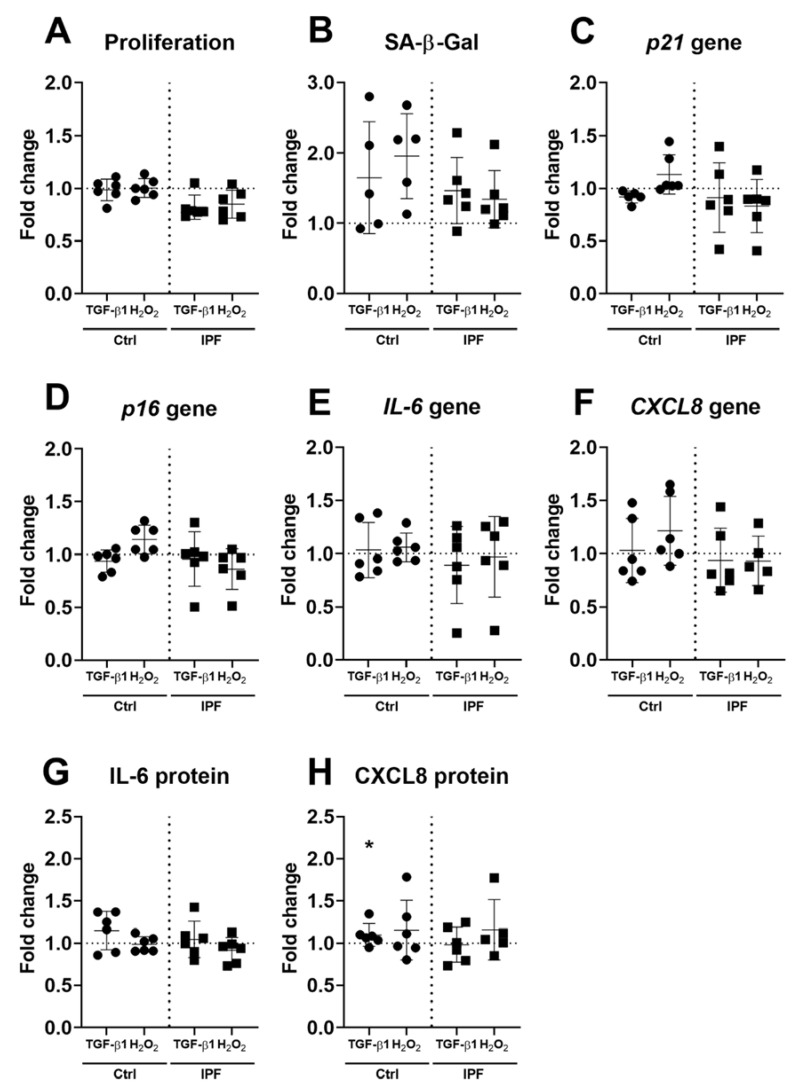
Markers of senescence in fibroblasts cultured on ECM derived from fibroblasts treated with either H_2_O_2_ or TGF-β1. Fibroblasts were cultured for three days on Ctrl- or IPF-derived ECM. Proliferation was assessed by cell enumeration (**A**), and SA-β-Gal-positive cells were counted after cytochemical staining (**B**). Relative levels of the cell-cycle inhibitors p21^Waf1/Cip1^ and p16^Ink4*a*^ (**C**,**D**). Panels (**E**–**H**) show gene expression and protein secretion of IL-6 and CXCL8 after three days of culture. Both gene expression and levels of cytokine production were normalised as described before and expressed as fold change to their respecting Ctrl– or IPF–ECM without treatment (*n* = 4–6). Data were analysed using repeated measures one-way ANOVA, or, if data points were missing, a mixed-effects analysis (REML), and considered significant at * *p* < 0.05.

**Figure 5 cells-10-01628-f005:**
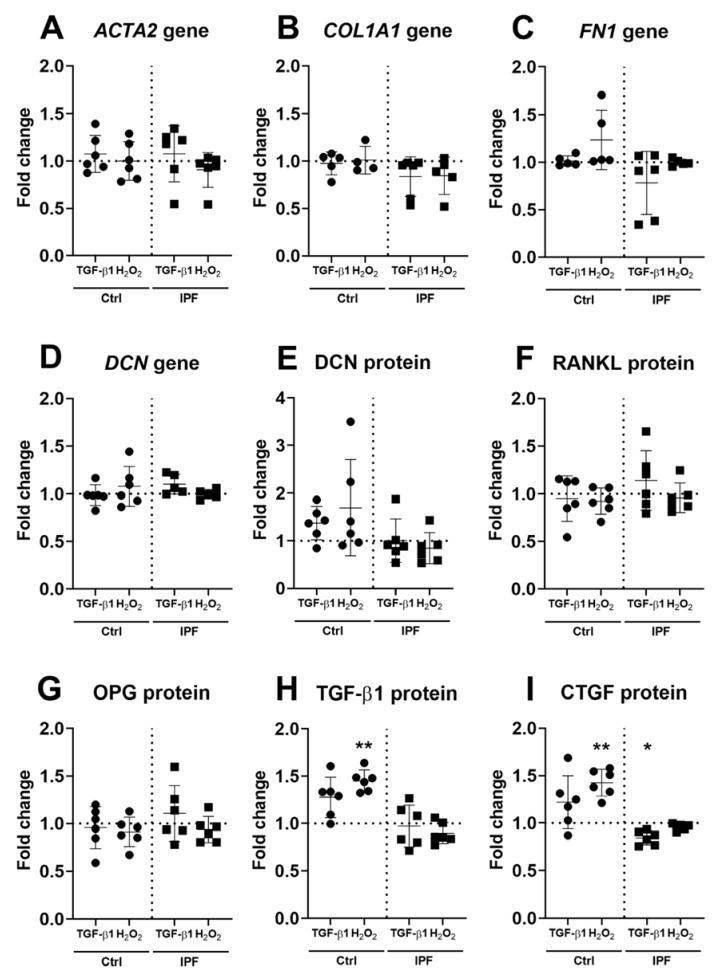
Fibrosis-associated gene expression and secretion of profibrotic cytokines. Fibroblasts were cultured for up to three days on treatment-derived Ctrl– or IPF–ECM before expression levels of ACTA2, COL1A1, FN1 and DCN were assessed (**A**–**D**). Panel (**E**–**I**) shows protein secretion of known fibrotic factors DCN, RANKL, OPG, TGF-β1 and CTGF after three days of culture. Both gene expression and levels of cytokine production were normalised as described before and expressed as fold change to their respective Ctrl– or IPF–ECM without treatment (*n* = 4–6). Data were analysed using repeated measures one-way ANOVA, or, if data points were missing, a mixed-effects analysis (REML). Significance shown as * *p* = 0.05 and ** *p* = 0.01 and data were considered significant at * *p* < 0.05.

**Table 1 cells-10-01628-t001:** Characteristics of fibroblasts donors for ECM substrate.

	Donor	Sex	Age	Smoking History	Pack Years
1	Ctrl-LF	M	59	N/A	N/A
2	Ctrl-LF	M	65	Ex	N/A
3	Ctrl-LF	M	69	Ex	20
4	Ctrl-LF	M	72	Smoker	55
5	Ctrl-LF	M	65	Ex	25
6	Ctrl-LF	M	60	Ex	20
7	IPF-LF	F	64	N/A	N/A
8	IPF-LF	M	67	N/A	N/A
9	IPF-LF	M	68	Never	N/A
10	IPF-LF	M	64	Ex	N/A
11	IPF-LF	M	61	Ex	36
12	IPF-LF	M	64	Ex	6

Ctrl-LFs = Control Lung Fibroblasts, IPF-LFs = IPF Lung Fibroblasts, F = Female, M = Male, Ex = Former smoker, N/A = Not available.

**Table 2 cells-10-01628-t002:** Characteristics of fibroblasts donors.

#	Donor	Sex	Age	Smoking History	Pack Years
1	Ctrl-LF	F	54	Current	38
2	Ctrl-LF	F	52	Current	40

Ctrl-LFs = Control Lung Fibroblasts, F = Female.

**Table 3 cells-10-01628-t003:** Primers and probed with designated exon boundaries.

Gene	Number	Exon Boundary
18S	Hs99999901_s1	1–1
CDKN2A	Hs00923894_m1	2–3
CDKN1A	Hs00355782_m1	2–3
IL-6	Hs00174131_m1	4–5
CXCL8	Hs00174103_m1	1–2
DCN	Hs00370385_m1	7–8
ACTA2	Hs00426835_g1	2–3
COL1A1	Hs00164004_m1	1–2
FN1	Hs01549976_m1	8–9
FBLN1C	Hs00242546_m1	14–15
TGF-β1	Hs00998133_m1	6–7
TGM2	Hs01096681_m1	9–10
LOX	Hs00942483_m1	5–6
LOXL1	Hs00935937_m1	6–7
LOXL2	Hs00158757_m1	10–11
LOXL3	Hs01046945_m1	4–5
LOXL4	Hs00260059_m1	14–15

18S ribosomal RNA = 18S, Cyclin-dependent kinase inhibitor 2A = CDKN2A, Interleukin 6 = IL-6, Chemokine ligand 8 = CXCL8, Decorin = DCN, Actin alpha 2 = ACTA2, Collagen 1α1 = COL1A1, Fibronectin = FN1, Fibulin-1C = FBLN1C, Transforming growth factor β1 = TGF-β1, Tissue transglutaminase = TGM2, Lysyl oxidase = LOX, Lysyl oxidase homolog 1–4 = LOXL1–4.

## Data Availability

The data presented in this study are available on request from the corresponding author.

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
