# Peer review of "Regulation of Cellular Senescence Is Independent from Profibrotic Fibroblast-Deposited ECM"

_cells, 2021, doi:10.3390/cells10071628_

Round 1

Reviewer 1 Report

In this interesting article, Blokland et al. investigated whether the extracellular matrix (ECM) secreted by lung fibroblasts derived from IPF patients promoted cellular senescence. They compared ECM derived from control and IPF fibroblasts at basal condition, in presence of  pro-fibrotic (TGFb1) or pro-senescence (H2O2) stimuli. In summary, the authors showed that IPF ECM did not induced fibroblast senescence but rather promoted myofibroblastic differentiation associated with pro-inflammatory and pro-fibrotic secretome. Overall, their findings are of interest, but some key results are not fully supported by the data.

Figure 3: the effect of IPF-ECM on ACTA2 mRNA expression is rather faint. The authors should confirm this finding by another method (western blot, immunofluorescence?). Does IPF-ECM also trigger morphological changes usually associated with myofibroblastic differentiation?

Figure 4 (panel G): the data (as displayed) are not really convincing.  IL-6 is supposed to be lower on H202-treated CTRL ECM but 3 samples are above the basal condition and 3 samples under it (a fold change around 1…). A similar comment can be done about the TGFb1-treated IPFL ECM. The authors should clarify this discrepancy.

Figure 5I: same comment with a fold change around 1 regarding conditions with statistical significance….

Reviewer 2 Report

qPCR:

What reference genes were used? Were the reference genes checked for consistent expression across conditions? Were the amplification efficiencies comparable? What was the LOQ for the primers? Were Ct values discarded because of too high Ct values? Were technical replicates measured and if yes, how were the data processed?

It’s not explained how dCt values were calculated. You show 2^(-dCt), so one might speculate that dCt = Ct[goi] – Ct[ref]. I recommend NOT transforming dCt values. dCt values are reasonably normal distributed, while assuming that 2^(-dCt) is normal distribution is unreasonable. See doi: 10.1049/enb.2017.0004 for further explanations.

Statistics:

What level of significance was used? If different levels were used, state the rationale.

I don’t understand the rationale behind the outlier testing. What scientific hypothesis is connected to the hypothesis tested with the ROUT test? What is the rational of choosing Q = 1%? And what if a value is signified as an outlier by this method? Is it removed from the analysis (and if so: with what justification?)? When the data were log-transformed for the subsequent analysis, was this transformed data used for the outlier test? Removed outliers should be reported (ideally even shown) in the figures. Ideally, the analysis should be done with and without excluding these “outliers”, and changes in conclusions should be discussed.

The data is from primary human cells, one should expect a huge variance. Removing outliers on an automated basis is likely introducing bias and over-confidence in the conclusions. It makes sense to check for vastly outlying values and then to cross-check if the measurements are invalid, highly implausible or even impossible, what would justify their removal.

It’s strange that the figure legend say you did signed-rank tests. This is not described in the methods and it does not seem applicable in all cases, as the numbers of values in the two groups are not always equal.

The figure legends don’t provide any information about how the data were analyzed. It’s not clear what variables were transformed. It’s unclear which p-values were corrected in which way for multiple testing. You wrote that you performed rmANOVAs and mixed models, but I don’t find any ANOVA table in the results or in the supplement.

In several places, you state that there is no difference between groups (e.g. Fig.2, Fig. 3). However, the data shown are consistent with considerable differences between the groups. A non-significant result of a hypothesis test is not indicating “no difference”. It only means that the data are inconclusive w.r.t. the tested hypothesis (if this is that the mean difference is zero, the data are inconclusive w.r.t. the sign of the difference). If you want to stress that the difference between the groups is negligible, you should instead give the confidence intervals of the differences and check if they are fully within the range of differences considered negligible (this is the simpler equivalent to “two-one-sided-t-tests”, TOST).

It's unclear to me what data refers to which fibroblast donor (you used two). Where they mixed? Is there any information about how the fibroblast "performance" differed between these two subjects?

3.3 / Fig. 3

You claim that ACTA2 gene expression is increased on IPF-ECM. The data don’t look convincing at all. Please state how this conclusion was derived. Please provide the data to follow this analysis. It’s not clear to me how the proteins were chosen. If the mRNA analysis shows a possible expression difference in a gene, that at least this gene should be analyzed on the protein level to see if the change can have any biological relevance.

3.4 / Fig. 4 and 3.5 / Fig. 5

Again, the data don’t look convincing, and the analysis is not explained in sufficient detail. Please provide the data and the details of the analysis.

Round 2

Reviewer 1 Report

The authors addressed almost all my comments and the manuscript is much improved. It is a pity that the editor did not give them enough time to address one of my major concerns (validation of ACTA2 data).